# Did I do that? Blame as a means to identify controlled effects in reinforcement learning

**Oriol Corcoll**                                                                *oriol.corcoll.andreu@ut.ee*
*Institute of Computer Science*
*University of Tartu*

**Youssef Mohamed**                                                          *youssef.mohamed@ut.ee*
*Institute of Computer Science*
*University of Tartu*

**Raul Vicente**                                                                  *raul.vicente.zafra@ut.ee*
*Institute of Computer Science*
*University of Tartu*

**Reviewed on OpenReview:** *https://openreview.net/forum?id=NL2L3XjVFx*

## Abstract

Affordance learning is a crucial ability of intelligent agents. This ability relies on understanding the different ways the environment can be controlled. Approaches encouraging RL agents to model controllable aspects of their environment have repeatedly achieved state-of-the-art results. Despite their success, these approaches have only been studied using generic tasks as a proxy but have not yet been evaluated in isolation. In this work, we study the problem of identifying controlled effects from a causal perspective. Humans compare counterfactual outcomes to assign a degree of blame to their actions. Following this idea, we propose Controlled Effect Network (CEN), a self-supervised method based on the causal concept of blame. CEN is evaluated in a wide range of environments against two state-of-the-art models, showing that it precisely identifies controlled effects.

## 1 Introduction

The recent success of reinforcement learning (RL) methods in complex environments such as Hide & Seek (Baker et al., 2019), StarCraft II (Vinyals et al., 2019), or Dota2 (OpenAI et al., 2019) has shown the potential of RL to learn complex behavior. Unfortunately, these methods also show RL's inefficiency to learn (Espeholt et al., 2018; Kapturowski et al., 2019; Gulcehre et al., 2020), requiring a vast amount of interactions with the environment before meaningful learning occurs. Consequently, environments with sparse rewards are known to be extremely difficult, making imperative a good exploration strategy. A popular approach to exploration is to introduce behavioral biases in the form of intrinsic motivators (Chentanez et al., 2005; Mohamed & Rezende, 2015). This technique aims to facilitate the learning of task-agnostic behavior by producing dense rewards, driving the agent to discover novel states and by doing so increase the chance of discovering the environment's reward.

Numerous motivators have been developed taking inspiration from humans, e.g. curiosity or control (Bellemare et al., 2012b; Pathak et al., 2017; Burda et al., 2018; Choi et al., 2019; Badia et al., 2020b). Recent work (Choi et al., 2019; Song et al., 2019; Badia et al., 2020a;b) has achieved State-of-the-Art on the Atari benchmark (Bellemare et al., 2012a) by rewarding agents for the discovery of novel ways to control their environment. A typical design principle among these methods is using an inverse dynamics model to predict the chosen action from two consecutive observations with the hope that the latent representation learned encloses aspects of the environment controlled by the agent. This approach assumes that the action can be recovered from consecutive observations, which is not always the case.

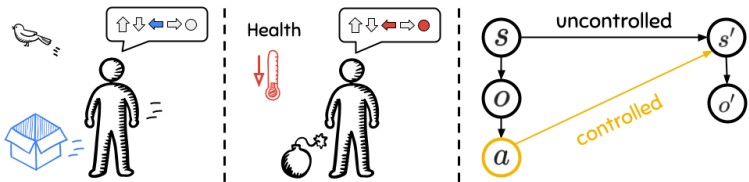

Figure 1: Left) Blame compares an imagined normative world to what actually happened to attribute the movement of the agent and the box to the action. Middle) Using a do-nothing action as normative world is not enough since do-nothing has an effect. Right) Controlled effects are the effects of an action on the environment.

A more causal approach is to compare counterfactual worlds (Pearl, 2009), i.e., an effect is controllable if the effect would have been different had the agent taken another action. A caveat of this approach is that things become trivially controllable. Fig. 1 (left) shows an scenario where an agent moves a box by moving left. Here the box becomes controllable even when the agent performs the action "do-nothing" since there is an action, "move-left", that would move the box. Contrarily, it is believed that humans identify controlled effects by assigning a degree of blame to their actions. In particular, humans compare what happened to what normally would happen (Halpern, 2016; Morris et al., 2018; Langenhoff et al., 2019; Grinfeld et al., 2020). If what happened is normal, humans would not assign blame to their actions, e.g. when performing "do-nothing", the box's effect would not be controlled since normally the box would not move. However, it would be considered controlled when moving left since its normative state is to not move.

This work proposes Controlled Effects Network (CEN), a completely unsupervised approach to identify controlled aspects of the environment based on the human notion of blame. CEN can be incorporated to any forward model and works by dividing its latent representation into two branches, a normal and a controlled view of the world. Our experiments show that CEN can disentangle effects precisely, outperforming state-of-the-art approaches to detect controlled effects.

## 2 Identifying controlled effects using blame

Our goal is to identify changes in the environment that were controlled by the agent. This section introduces Individual Causal Effect (ICE), a fundamental measure in causal literature, and frames it in the context of RL. We show how this measure can be used to identify controllable effects but argue that these are not suitable for RL. In contrast, the human perception of causality is associated with the concept of blame (Gerstenberg & Lagnado, 2014). For example, if lightning hits a forest tree and starts a fire, humans would point to the lightning as the cause of the fire, not the oxygen or wood since they are normally present in the forest. Consequently, we expand the idea of blame to identify controlled effects by using measures of normality and counterfactuals.

### 2.1 Controllable effects

What does it mean to cause something? Pearl et al. (2016) provide an intuitive definition of cause-effect relations: "A variable X is a cause of a variable Y if Y, in any way, relies on X for its value". This kind of formulation tries to answer questions like "does smoking causes cancer?". Actual causality, proposed in Halpern (2016), studies causal relations between individual events of $X$ and $Y$. It aims to answer questions like, "did smoking for 30 years caused David's cancer?". The individual nature of Actual causality makes this framework especially suited to RL problems. In the following, we introduce the concept of causal effect in the context of RL.

The individual causal effect (ICE) of an event $X = x$ on a variable $Y_i$ can be measured by comparing counterfactual worlds

$$ICE_{Y_i}^x \equiv Y_i^x - Y_i^{\tilde{x}} \; , \tag{1}$$

where $Y_i^x$ reads as "what would the value of an individual $Y_i$ be if $X$ is forced to be $x$". Similarly, $Y_i^{\tilde{x}}$ describes the value of $Y_i$ when $X$ is forced to not be $x$. Note that the sub-index $i$ refers to an individual, and hence in the following, we use $Y_i^x$ and $Y^x$ interchangeably.

The *fundamental problem of causal inference* states that we can only observe one of these counterfactual worlds and the other needs to be imagined. Intuitively, Eq. 1 compares the world where the event $x$ happened to an alternative world where event $x$ had not happened. Consequently, we say that $x$ has a causal effect on $Y$ if there is an $\tilde{x} \in X$ that makes Eq. 1 nonzero. In the context of RL, $X$ and $Y$ take the form of actions, states and/or observations. Fig. 1 (right) illustrates the causal relations present in a typical RL setting, where a state $s$ has an effect on both the next state $s'$ and the produced observation $o$ which, in turn, has an effect on the agent's choice of action $a \in \mathcal{A}$. Similarly, an action has an effect on the next state. Since states are typically not accessible by the agent, we do not use states as variables; nevertheless the same principle can be applied. We define the perceived effect $e_p^a$ as the difference between consecutive observations when taking action $a$, i.e. $e_p^a \equiv o' - o$. As in Eq. 1, we say that a perceived effect is controllable by the agent's action when

$$\exists \tilde{a} \in \mathcal{A}\colon \left(e_p^a - e_p^{\tilde{a}}\right) \neq 0 \ . \tag{2}$$

Since we want to know what elements of the perceived effect are controllable, the difference is an element-wise operation. It is important to notice that Eq. 2 has far-reaching consequences, for example, an agent next to a box would have a causal effect on it even when doing nothing since there is a counterfactual world where that box would have moved. If we use Eq. 2 as reward, the agent would be rewarded for almost every action at every state! Note that taking $\tilde{a}$ as a special "do-nothing" action would not work since even doing nothing does something, e.g., Fig. 1 (middle) shows a scenario where doing nothing has an effect on the agent's health. Taking do-nothing as $\tilde{a}$ would not attribute the effect to the agent. Instead, we would want a more human-like definition of what is controlled where an agent controls a box if moved or its life if a bomb could have been easily avoided.

## 2.2   Blame

It has been shown that the human notion of causality is affected by what is normal (Kahneman & Miller, 1986; Cushman et al., 2008; Knobe & Fraser, 2008; Hitchcock & Knobe, 2009). Here, we resort to concepts of normality from actual causality to find if the agent's action is to blame for what happened. Halpern & Hitchcock (2014) propose to compare what actually happened with what normally would happen. Following this idea we build a normative world to replace $Y^{\tilde{x}}$ in Eq. 1

$$ICE_Y^x = Y^x - \beta_Y \ , \tag{3}$$

where $\beta_Y$ is the value $Y$ would normally take. Such a value is of course contingent to the notion of normality used, which is for us to define. We can reformulate Eq. 3 to compute the controlled effect of an action $e_c^a$ as

$$
\begin{aligned}
e_c^a = ICE_{e_p}^a = e_p^a - \beta_{e_p} \\
= e_p^a - \mathop{\mathbb{E}}_{\tilde{a}, o'}\left[e_p^{\tilde{a}}\right] \ ,
\end{aligned}
\tag{4}
$$

where $\beta_{e_p}$ represents what normally would have happened to the environment. This quantity not only depends on the environment's dynamics but also on the agent's typical behavior.

In this work, we compute $\beta_{e_p}$ as the expectation over all possible futures $o'$ of each action in $\mathcal{A}$. Note that stochastic environments may have be multiple next observations for each action. To simplify notation, the following sections use normal effect as $e_n = \mathbb{E}_{\tilde{a}, o'}\left[e_p^{\tilde{a}}\right]$. Intuitively, Eq. 4 builds a normal world by observing every alternative $e_p$ produced by each action creating an average perceived effect. Consider the example in Fig. 1 (middle), moving left or doing nothing would make the agent's health decrease. Eq. 4 would indicate that what is normal is for health to not change since in average health stays the same; thus, the loss of health when moving left or staying would be attributed to the agent. On the other hand, moving right would only attribute the change in the agent's position as controlled. Note that the explosion would never be credited to the agent since it would have happened no matter what action is taken.

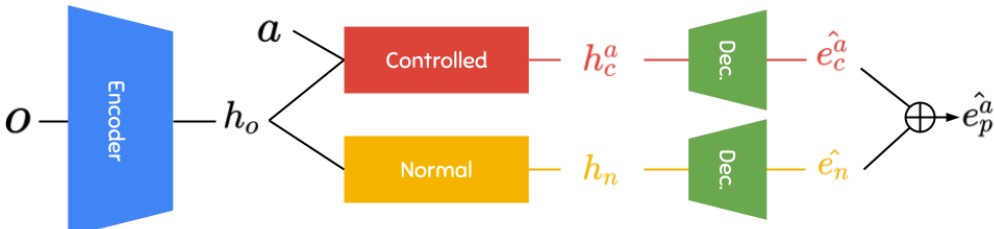

Figure 2: CEN divides the latent space of a forward model into controlled and normal branches. Each branch disentangles controlled and normal effects and decodes each into pixel space independently.

Special care needs to be taken when constructing the normal world $\beta_{e_p}$ for continuous or large action spaces. Computing counterfactuals on an infinite number of possibilities cannot be done and some approximation needs to be implemented. Although our experiments use discrete actions, the proposed method in the following section is equipped to handle continuous action spaces since it does not compute counterfactuals for each possible action but approximates the normal world directly. It is also important to notice that the controlled effects that Eq. 4 can identify in a partially observable setting ($o \neq s$) are constrained to those observed by the agent. Nevertheless, humans cannot perceive every change in state but can identify relevant controlled effects for their survival and joy.

## 3 Unsupervised learning of controlled effects

In practice, we do not have access to every world and cannot compute Eq. 4 directly. We propose **Controlled Effects Network (CEN)**, an unsupervised method depicted in Fig. 2 that disentangles controlled and normal effects only using perceived effects as a self-supervised training signal. CEN is modeled as a neural network and it is based on a forward model, where observation and action are used to predict the outcome of performing such action on the environment. In contrast to conventional forward models, CEN divides its latent space into controlled and normal representations; similarly to Dueling Networks (Wang et al., 2015). These two representations approximate the controlled and normal effects in latent space. A decoder converts these latent representations into pixel space allowing to estimate $e_c^a + e_n = e_p^a$ as in Eq. 4.

The **controlled branch** has privileged access to the action, consequently, having only this branch would make CEN a regular forward model, i.e., the controlled branch alone can predict the perceived effect resulting from the action. Then, why do we need the **normal branch**? The role of the normal branch is to force the controlled branch to predict only what is not predictable from the observation alone and hence, modeling what is controlled by the agent. In a way, the normal branch acts as a distillation mechanism where only what can be controlled will be represented by the controlled branch. To promote the controlled branch to model only controlled effects, we use the following objective

$$\mathcal{L} = \text{MSE}\left(\hat{e_c^a} + \hat{e_n}, e_p^a\right) + \alpha \ \text{MSE}\left(\hat{e_n}, e_p^a\right), \tag{5}$$

where the first term is the reconstruction loss in which the predicted target is compared to the perceived effects provided by the environment. The second part of the loss enforces the network to use the normal branch to model the world as much as possible. Since this branch cannot predict everything without the action the model will converge to the expected effect due to the MSE loss. Additionally, a hyperparameter $\alpha$ regulates how much the normal branch should model the environment. In practice, we found that this hyperparameter creates an agreement between branches on uncertain futures which seemed to be critical in environments with stochastic entities. Note that we combine normal and controlled effects in pixel space, this is because we would like to know the pixels that are controlled by the agent. Nevertheless, if the downstream task CEN is part of only needs the latent representation, the combination of controlled and normal representations could be done in latent space i.e. $\hat{h_c^a} + \hat{h_n}$.

Let us look again at the example in Fig. 1 (middle) and assume the agent picks the do-nothing action. The normal branch is encouraged to model the bomb since it does not depend on the action. Furthermore, it

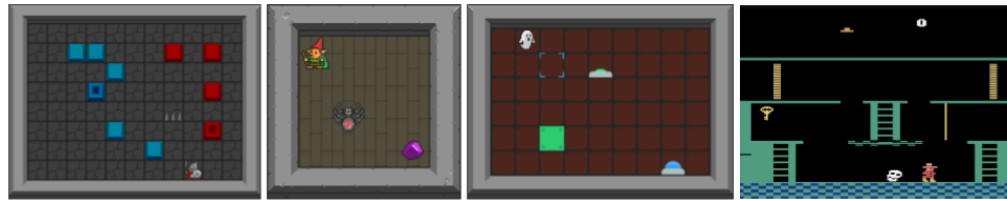

Figure 3: Suite of environments used for the experiments. From left/right top/bottom: Clusters, Lights, Spiders and Montezuma's Revenge (MZR).

should not predict any change in health since what is normal is for health to not change. Thus, the controlled branch must model the change in health.

# 4 Experiments

This section evaluates CEN[1] on the following main questions: 1) Can CEN identify controlled effects at pixel-level? i.e. can it produce an accurate segmentation mask? 2) Some applications may not require pixel-level precision; we asses CEN on predicting controlled effects at attribute-level from both pixel masks and latent representations. Every experiment reports a 95% confidence interval[2] using three different seeds.

**Environments:** we use multiple environments (Fig. 3) to answer the above questions, each showcasing a different aspect of what can be controlled. These environments are based on Griddly (Bamford et al., 2020) and Atari ALE (Bellemare et al., 2012a), both using the Gym interface (Brockman et al., 2016). More details of these environments are given in each experiment and appendix.

**Baselines:** We use Attentive Dynamics Model (ADM) (Choi et al., 2019) in experiments 1) and 2). ADM is an action-prediction model based on an spatial attention mechanism. It works by predicting the action performed on individual image patches. Then, a spatial attention mechanism selects a sparse set of patches to use when making a final prediction of the action. The masks produced by the attention mechanism are considered controlled aspects of the environment. Although ADM does not produce pixel level masks, only at patch level, to our knowledge ADM is the most competitive method to provide pixel-level information about controlled aspects. For 2) we rely on the inverse modeled proposed in Never Give Up (NGU) (Badia et al., 2020b), the current SOTA for exploration in RL.

**Implementation:** CEN is implemented as an encoder-decoder architecture with 2D convolutional layers and ReLU activation functions; the normal and controlled branches are implemented with linear layers. Additionally decoder weights are shared. Throughout the experiments we use the same neural networks and hyperparameters unless specified otherwise. Our implementation of ADM uses the same architecture and hyperparameters proposed in Choi et al. (2019). See appendix for more details on the architecture and hyperparameters.

## 4.1 Controlled effects at pixel-level

This set of experiments explores CEN's ability to identify pixels corresponding to controlled entities. Although CEN computes the magnitude and direction of the effects, we create a binary mask by setting a threshold for the predicted controlled effects (see exact details in appendix D). We report pixel F1 scores between ground truth and predicted binary masks. We explain how ground truth masks are computed below, but it should be clear that CEN is a fully unsupervised method that does not use this ground truth in any way, ground truth is only used to evaluate the produced masks. The network is trained to minimize Eq. 5 using the ADAM optimizer (Kingma & Ba, 2015) and 500K samples of the form $(o, a, e_p^a)$ collected using a random policy.

---

[1] Networks, training and evaluation have been implemented using PyTorch (Paszke et al., 2019), NumPy (Harris et al., 2020) and PFRL (Fujita et al., 2021); our experiments are managed using W&B (Biewald, 2020).

[2] CI are computed using bootstrap resampling as per the Seaborn (Waskom, 2021) package

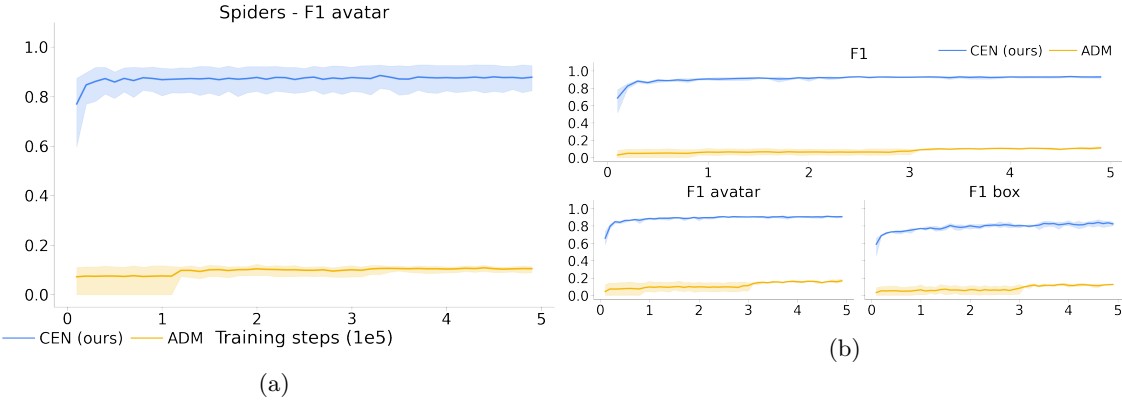

(a)

(b)

Figure 4: a) CEN can correctly disentangle the agent from the randomly moving objects. b) Clusters environment where CEN is able to model not just the agent but also the movement of boxes.

**Ground truth masks:** Griddly provides access to each entity's state (x, y, light is on/off, etc); a binary mask is produced for each controlled entity with any of its attributes changed when transitioning between two time steps. x and y coordinates are projected into pixel space and a bounding box is generated using the size of that entity. The resulting masks for each entity are combined into a single mask $m$ by taking the maximum value among them. Since we want to know what pixels were actually controlled, the final ground truth mask is produced as: $m \cdot e_a^p$.

In case of MZR, Atari ALE allows us to actually compute counterfactual worlds by saving and loading the state of the game (RAM) multiple times when taking different actions. For this, we directly compute Eq. 4 using ALE's special calls *cloneSystemState* and *restoreSystemState*. More precisely, we compute every possible perceived effect reachable from the current state and build a normal effect using the mode over all possible effects. Then, we compare the perceived effect for the agent's chosen action against the normal effect. The ground truth mask will have 1s where these two effects are different.

**CEN and ADM masks:** CEN's controlled masks are generated using the encoder, controlled branch and decoder. The predicted controlled effect is binarized using a threshold $T$ as $(-T < \hat{e}_c^a)|(e_c^{\hat{a}}c > T)$. In the case of ADM, its attention mask is thresholded in the same way and the mask is resized to the size of the effect.

### 4.1.1 Controlled vs uncontrolled effects

Here we use the Spiders environment to evaluate CEN's ability to disentangle controlled from uncontrolled effects. This environment has two main entities, the agent and a spider. The controlled masks must only focus on the agent and ignore the spider.

Fig. 4a shows the pixel F1 score for our model and the baseline. CEN is able to correctly disentangle controlled effects and can produce accurate masks. Although our implementation of ADM can predict the agent's action with 88% accuracy, it is not capable of modeling the agent's controlled pixels. We conjecture that this is due to ADM's sparse softmax mechanism; nonetheless this behavior persisted when increasing its entropy weight which should produce more dense masks.

### 4.1.2 Nearby controlled effects

Models based on action prediction are expected to work well on aspects related to the agent. For example, if an agent moves a box due to moving right; the box's movement is also controlled. It is unclear why these models would pay attention to the box since just knowing where the agent is, suffices to predict the chosen action. CEN's controlled branch, on the other hand, is motivated to model the box's effect since the normal branch would predict that the box stays where it is. We call "nearby" controlled effect to an effect that happens adjacent the agent, like the box's movement. To evaluate CEN on this kind of effects we use the Clusters environment where an agent needs to move colored boxes to their corresponding fixed colored blocks.

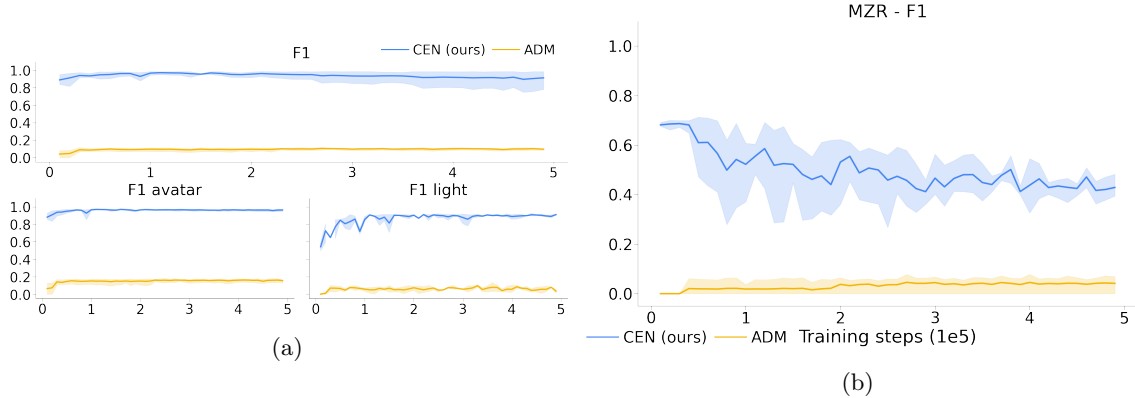

Figure 5: a) F1 score on the Lights, and b) MZR environments. CEN outperforms the baseline even in an environment with more complex dynamics and features.

Fig. 4b shows that CEN can precisely model effects on the agent and boxes. We breakdown individual effects to account for the class imbalance between the agent's movement and the boxes. CEN seems to make more mistakes with boxes than the agent but nonetheless, it can consistently model both.

### 4.1.3   Faraway controlled effects

In contrast to the previous experiment, we want to evaluate if CEN can model distant effects, i.e. effects that are reasonably far away from the agent's location. In this case, we use the Lights environment. Here the environment presents two buttons of different color that, when pressed, turn on their corresponding light. Lights are relatively far away from their corresponding buttons thus making it difficult to model them. As show in Fig. 5a, CEN is able to model this kind of effects.

### 4.1.4   Controlled effects in Montezuma's Revenge

This last pixel-level experiment evaluates CEN on Montezuma's Revenge (MZR) environment. Although agents in Atari environments have limited control over the environment, the relatively complex dynamics of Montezuma's Revenge makes it a challenging test-bed. Results shown in Fig. 5b indicate that CEN can also model controlled effects.

Unlike previous environments, CEN's F1 score behaves differently due to two main reasons: 1) *F1 decreases over time:* the F1 score is extremely sensitive for empty ground truth, a single pixel marked as controlled takes the score from 1 to 0. At the beginning of training, CEN produces empty masks correctly but performs poorly at controlled effects, which makes the F1 score artificially high. Over time it improves at controlled effects but starts marking very few pixels as controlled for effects where nothing is controlled. 2) *F1 is lower:* the sensitivity of the F1 score also affects CEN's performance on controlled effects since pixels outside the ground truth are penalized the same no matter how close to the ground truth they are. Appendix B.3 includes additional masks showing that CEN fails by pixels close to the controlled object instead of identifying the wrong object as controlled.

### 4.2   Controlled effects at attribute-level

In some cases requiring pixel-level precision can be excessive. The following experiments analyze how different representations can predict effects on attributes from the environment's state, e.g. changes on the agent's $(x, y)$ location or if a light turned on. To this end, we use a probing technique (Alain & Bengio, 2016) similar to the one described in Anand et al. (2019). This approach trains a classifier per each attribute of interest in the environment's state using frozen versions of trained networks to produce the classifiers inputs. More specifically we use two different sources for the probing classifiers, pixel-level masks (as in the previous section) or the model's latent representation.

|  |  | F1 Pixels | | F1 Latent | |
|---|---|---|---|---|---|
| Environment | Attribute | CEN (ours) | ADM | CEN (ours) | Inverse |
| Spiders | Agent | 1.0±0.00 | 0.47±0.23 | 0.97±0.01 | 0.67±0.05 |
|  | Spider ↓ | 0.35±0.03 | 0.25±0.03 | 0.41±0.01 | 0.44±0.02 |
| Clusters | Agent | 0.76±0.41 | 0.28±0.08 | 0.97±0.01 | 0.56±0.09 |
|  | Box | 0.78±0.37 | 0.32±0.19 | 0.95±0.02 | 0.77±0.00 |
| Lights | Agent | 0.97±0.01 | 0.33±0.15 | 1.0±0.01 | 0.84±0.08 |
|  | Button | 0.93±0.05 | 0.33±0.01 | 0.99±0.0 | 0.99±0.0 |
|  | Light | 0.93±0.04 | 0.41±0.14 | 1.0±0.0 | 0.99±0.0 |
| MZR | Agent | 0.66±0.08 | 0.42±0.23 | 0.91±0.02 | 0.88±0.02 |
|  | Skull ↓ | 0.19±0.03 | 0.20±0.08 | 0.61±0.03 | 0.61±0.04 |

Table 1: F1 score when predicting attributes from pixel or latent space. Lower is better for Skull and Spider.

For the first case, we produce a binary mask, as in the previous experiments, to occlude perceived effects and use these to train each probing classifier. Classifiers have to predict if there was a positive, negative or none effect. Note that the classifiers need to predict any effect, not just controlled. Thus, the classifier should only be able to predict accurately controlled attributes such as the agent's position but should fail at predicting uncontrollable effects like the location of the spider or skull. We use a random policy to collect a dataset of 35K samples of the form $(m * e_p, y)$ where $m$ is the mask produced by the model and $y$ is the ground truth class. Each dataset is split into a typical 70/20/10, allowing a 20% class imbalance. We report F1 score of each attribute on the test set.

### 4.2.1 Attribute probing from pixels

The results in Table 1(first block) indicate that CEN consistently outperforms the baseline when predicting controlled effects for state's attributes, and thus modeling controlled effects accurately. Furthermore, for both Spiders and Montezuma's Revenge environments the model cannot predict the uncontrolled effects, as expected. Even though ADM's action prediction accuracy was high ($\sim 88\%$) on every environment, it is not able to consistently predict controlled effects at attribute-level.

### 4.2.2 Attribute probing from latent representations

In this case, we train classifiers using a latent representation instead of pixels. We use the latent representation from CEN's controlled branch ($h_c$). It is unclear how to create a latent representation from ADM, so we use an inverse model. The features of current and next observations are concatenated to create a latent

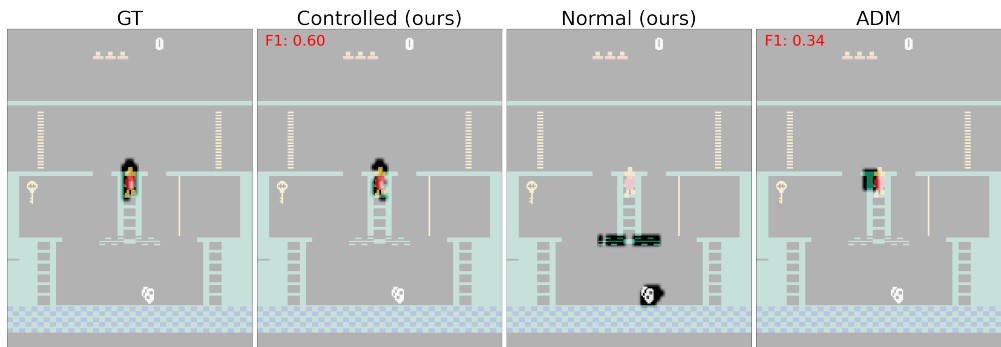

Figure 6: Example masks for Montezuma's Revenge. Additional masks are included in appendix B.3.

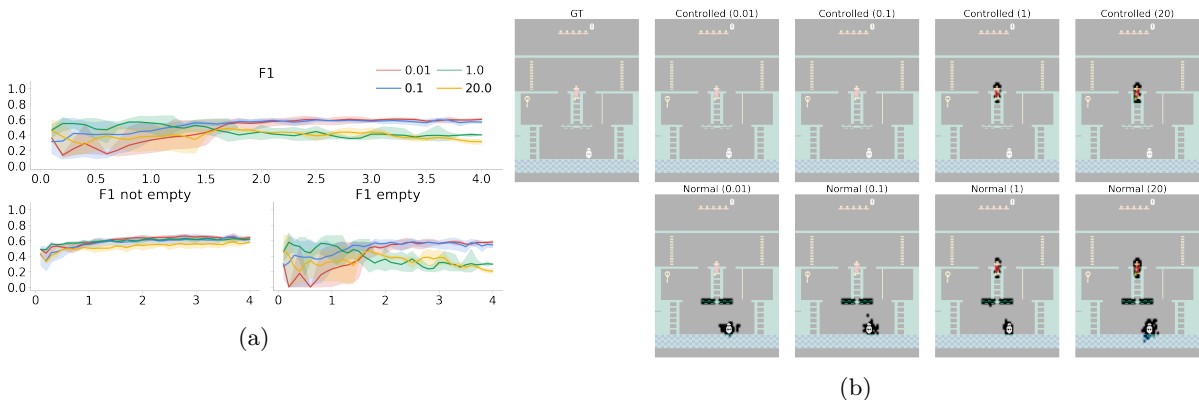

Figure 7: Ablation study on the effect of $\alpha$ from Eq. 5 on the learned representations of the normal branch. X axis of a) indicate training steps (1e5)

representation of what is controlled. As before, we train a linear probe to predict controlled attributes from the latent space of these models. The probe predicts changes on the x, y and direction (if applies) for agent, spider, skull and boxes; and on/off for lights and buttons.

Table 1 (second block) show that CEN improves on the baseline's performance. Although the inverse model is closer to CEN's performance than ADM, it still has difficulties predicting the agent and box changes in location. The reason of having high score in Skull is that the Skull is still only when the agent dies, making it easy to predict from the agent's position. Removing this event leads to a score of $\sim 0.35$ for both models.

### 4.3 How does the alpha hyperparameter affects CEN?

Here we study how $\alpha$ affects the loss in Eq. 5. This experiment uses Montezuma's Revenge and the same setup as in Experiment 4.1. We analyze the effect of alpha on CEN using values ranging from 0.01 to 20.

As can be seen in Fig. 7a, the different alphas do not impact the performance when the ground truth or controlled effects are not empty. On the other hand, the performance degrades the higher the alpha. We hypothesize that this behavior happens when the normal branch is forced to model controlled effects too strongly and the controlled branch needs to counter those bad predictions. In this case, the controlled branch will produce wrong masks, especially for empty ground truth. Fig. 7b shows that the higher the $\alpha$ the more the controlled branch needs to remove pixels modeled by the normal branch.

## 5 Related work

**Intrinsic motivators:** A popular way of introducing behavioral biases in RL agents is the use of intrinsic motivators (Singh et al., 2005; Mohamed & Rezende, 2015). These motivators can promote different types of exploration, from observational surprise (Burda et al., 2018) to control seeking agents (Pathak et al., 2017; Choi et al., 2019). Methods in the latter category have shown extremely good results achieving SOTA in important benchmarks. Choi et al. (2019) proposed Attentive Dynamics Model (ADM), an attention based method that discovers controlled elements in the environment and rewards the agent for discovering them. This method and its extension (Song et al., 2019) showed SOTA in Montezuma's Revenge. Badia et al. (2020b) combined control and observational surprise to promote exploration. Their method uses an episodic memory with an inverse model to promote the discovery of controlled effects and Random Network Distillation (Burda et al., 2018) to promote long term progress; again achieving SOTA in Atari's hard exploration environments. These methods show the importance of identifying what an agent can control.

**Causality in deep reinforcement learning:** Causality is central to humans; we think in terms of cause-effect. A similar method to Blame was proposed in Chattopadhyay et al. (2019), where they use causal attribution methods to analyze the effect of inputs on a neural network's outputs. Recent work has introduced

causality into deep reinforcement learning (Schölkopf et al., 2021; Ke et al., 2021; Foerster et al., 2018; Buesing et al., 2018; Jaques et al., 2018; Dasgupta et al., 2019; Goyal et al., 2019; Nair et al., 2019; Madumal et al., 2020) showing that this is a promising avenue for the training of agents. Corcoll & Vicente (2020) proposed an attribution method to learn temporal abstractions for object-centric hierarchical RL. Bellemare et al. (2012b) compute controllable aspects of the environment by generating a mask with all possible controllable areas of an image and uses it as part of the policy's input. In this work, we identify the controlled effects of individual actions using causal concepts of normality and blame. A similar concept to normality has been explored in control theory by Todorov (2009) called "passive dynamics". Passive dynamics compute how a dynamical system would evolve without interventions.

## 6 Conclusions and limitations

This work provides a causal perspective to the problem of identifying controlled aspects of the environment and proposes Controlled Effect Network (CEN), a fully unsupervised approach to this problem. CEN creates a normative world using counterfactuals and compares what actually happened with what normally would happen to attribute changes on the environment to the agent. The presented experiments show that, despite being unsupervised, this method precisely identifies controlled effects. In future work, we will explore CEN as intrinsic motivator or as a way to discover skill in a hierarchical reinforcement learning setting.

### 6.1 Limitations

*Normality:* although we propose a measure of normality in Eq. 4, this is far from ideal. For example, the resulting normal world may not be real/possible under the environment dynamics, a more realistic solution would be to use the mode instead of expectation. This could be achieved by using a critic, as in GANs (Goodfellow et al., 2014). We believe the way humans see normality is context dependent and should be learned instead of a fixed function. This is an active research area in the field of psychology where researchers look to underpin human causal judgment.

*State instead of observations:* our current formulation of ICE uses perceived effects as opposed to states. When CEN indicates that some change is controlled this is not (necessarily) equivalent to stating that the objects represented by those pixels are controlled. An exciting avenue to explore is combining CEN with a state representation model (e.g. LSTM or RSSM by Hafner et al. (2019)).

*Reliance on policy:* since Eq. 4 does not depend on the policy, CEN needs diverse data for each action to approximate it, ideally from a random policy. This important problem is not specific to CEN, any forward model needs a policy that explores multiple actions to provide accurate predictions.

*Multi-step controllability:* CEN assumes that objects are controlled immediately after performing an action. A valuable extension is to identify the consequences of an action multiple steps ahead. Possibly by incorporating Blame to works like Mohamed & Rezende (2015); Gregor et al. (2016).

### Acknowledgments

The authors would like to thank Jaan Aru and Daniel Majoral for insightful comments on the manuscript. This work was supported by the University of Tartu ASTRA Project PER ASPERA, financed by the European Regional Development Fund.

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

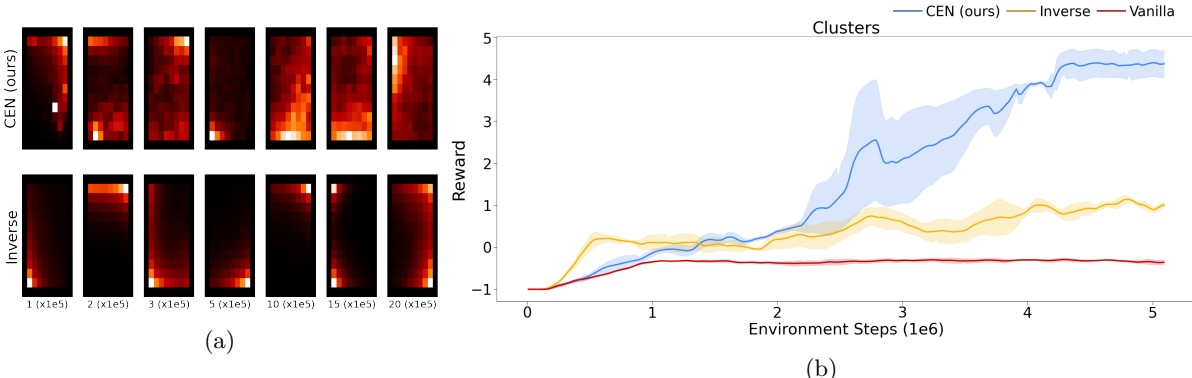

Figure 8: a) State visitation maps at different points of training of the Empty environment. CEN values different locations similarly, and consequently, the agent learns to explore states more uniformly. The inverse model encourages going to walls where predicting the action is hard. b) CEN promotes the movement of boxes and consequently faster learning

# A Appendix

# B Additional experiments

## B.1 CEN as intrinsic motivator

We have shown that CEN can learn controlled effects in an unsupervised manner. Here we showcase the use of this ability as an intrinsic motivator of a reinforcement learning agent. We consider two tasks, an empty environment without any extrinsic reward where the agent can only control itself and Clusters. The RL agent is implemented using PPO (vanilla). Additionally, PPO is augmented with the exploration bonus proposed in Never Give Up (NGU + Inverse) (Badia et al., 2020b). NGU is composed of two modules for computing episodic and life-long rewards. For simplicity the following experiments only use the episodic module which in NGU consists of a count-based method using an episodic memory to approximate the number of times an agent visited each state, and an inverse model to identify controlled states. We replace the inverse model with CEN (NGU + CEN) and use the latent representation of the controlled branch to compute NGU's episodic reward. In this experiments, CEN is trained along the PPO policy using experiences collected with the same policy.

**Empty environment:** the goal of this environment is to showcase how each NGU variant rewards controlled events. In this environment, only the agent's movement is controlled; thus, rewarding for controlling the agent's location should promote a uniform exploration of the environment since once a location is visited, it should not be rewarded as much as visiting a new location. We hypothesize that an inverse model will create similar representations for the same action disregarding the location where it was taken. This should impair exploration since the reward will be similar regardless of where the action is performed.

Fig. 8a shows that agent trained with the inverse model hogs walls and corners. This is because is hard to predict the action near unmovable objects, leading to higher value. Contrarily, CEN promotes different locations more uniformly and the agent learns to explore better the environment.

**Clusters environment:** A more challenging environment is Clusters, where the agent needs to move colored boxes to their respective colored blocks. This environment provides a reward at the end of the episode corresponding to the total number of boxes correctly placed. Results are provided in Fig. 8b. Due to the sparsity of the reward PPO does not learn a correct behavior in the given time. Similarly, NGU + Inverse learns to place one box but fails to learn a general behavior to solve the task. Conversely, NGU + CEN quickly learns to move boxes leading to a high extrinsic reward.

## B.2 Additional baselines for attribute prediction

The following table is an extension of Tab. 1 with additional simpler baselines. The **No mask** baseline does not filter any pixel from the image, i.e., the mask is all ones. We would expect close to perfect prediction of the attributes since all the information is in the input of the probing network, our results confirm that this is the case.

The **CEN (rand.)** and **Normal (rand.)** baselines are untrained, randomly initialized, CEN networks where one provides the controlled hidden representation and the other the normal hidden representation to the probing network. This experiment aims to isolate the benefits of the architectural biases introduced by CEN.

Interestingly, when there is no uncontrolled effects (Clusters and Lights), the controlled branch provides a good representation. Even with random weights the network is able to encode the action in the representation, making it easier for the linear probe to predict the agent attribute. Action and agent are correlated making the task of the linear probe easier. If the relation between object and action is more complex (e.g. boxes, lights or buttons), the linear probe has more difficulties. This indicates that the architectural biases are enough for simple relations but the more complex the relation the more learning is needed. When there are uncontrolled effects (Spiders and MZR) the representation is entangled and the performance of the linear probe is low.

| | | F1 Pixels | | F1 Latent | | |
| --- | --- | --- | --- | --- | --- | --- |
| Environment | Attribute | CEN (ours) | No mask | CEN (ours) | CEN (rand.) | Normal (rand.) |
| Spiders | Agent | 1.0±0.00 | 1.0±0.00 | 0.97±0.01 | 0.56±0.06 | 0.16±0.01 |
| | Spider ↓ | 0.35±0.03 | 1.0±0.00 | 0.41±0.01 | 0.29±0.01 | 0.17±0.01 |
| Clusters | Agent | 0.76±0.41 | 1.0±0.00 | 0.97±0.01 | 0.95±0.01 | 0.17±0.01 |
| | Box | 0.78±0.37 | 1.0±0.00 | 0.95±0.02 | 0.84±0.04 | 0.20±0.02 |
| Lights | Agent | 0.97±0.01 | 1.0±0.00 | 1.0±0.01 | 0.95±0.01 | 0.18±0.01 |
| | Button | 0.93±0.05 | 1.0±0.00 | 0.99±0.0 | 0.59±0.04 | 0.49±0.14 |
| | Light | 0.93±0.04 | 1.0±0.00 | 1.0±0.0 | 0.56±0.06 | 0.47±0.05 |
| MZR | Agent | 0.66±0.08 | 0.97±0.02 | 0.91±0.02 | 0.62±0.02 | 0.17±0.01 |
| | Skull ↓ | 0.19±0.03 | 1.0±0.01 | 0.61±0.03 | 0.48±0.02 | 0.36±0.00 |

Table 2: F1 score for CEN and additional baselines when predicting attributes from pixel or latent space.

## B.3 Masks

Here we provide masks for both CEN and ADM. Mask are based on effects and extend over two frames, for visualization purposes here we only show the next observation with the full mask. It can be seen that CEN fails for a few pixels in MZR, which may explain the low F1 score for empty GT.

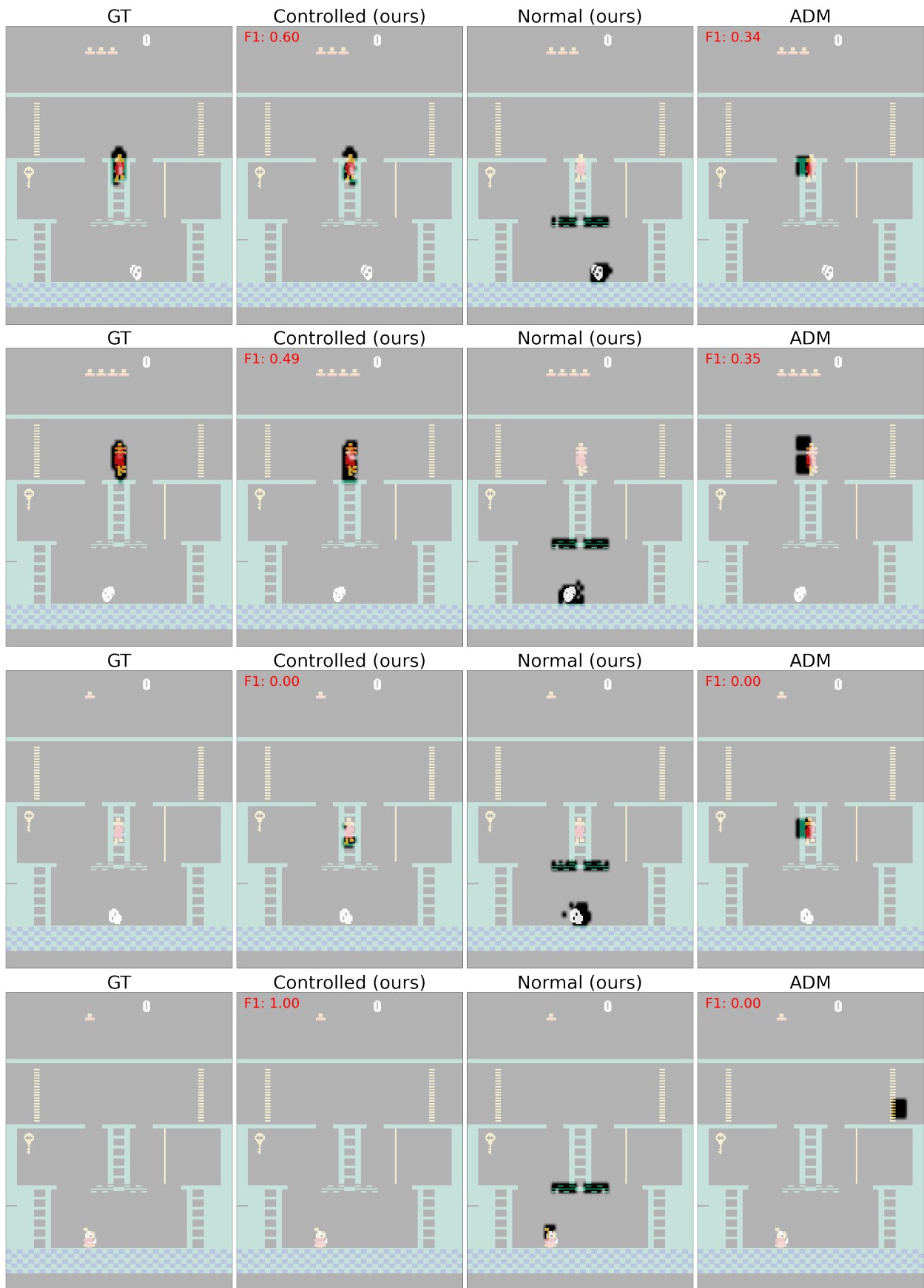

Figure 9: Examples of success and failure cases for CEN and ADM in MZR.

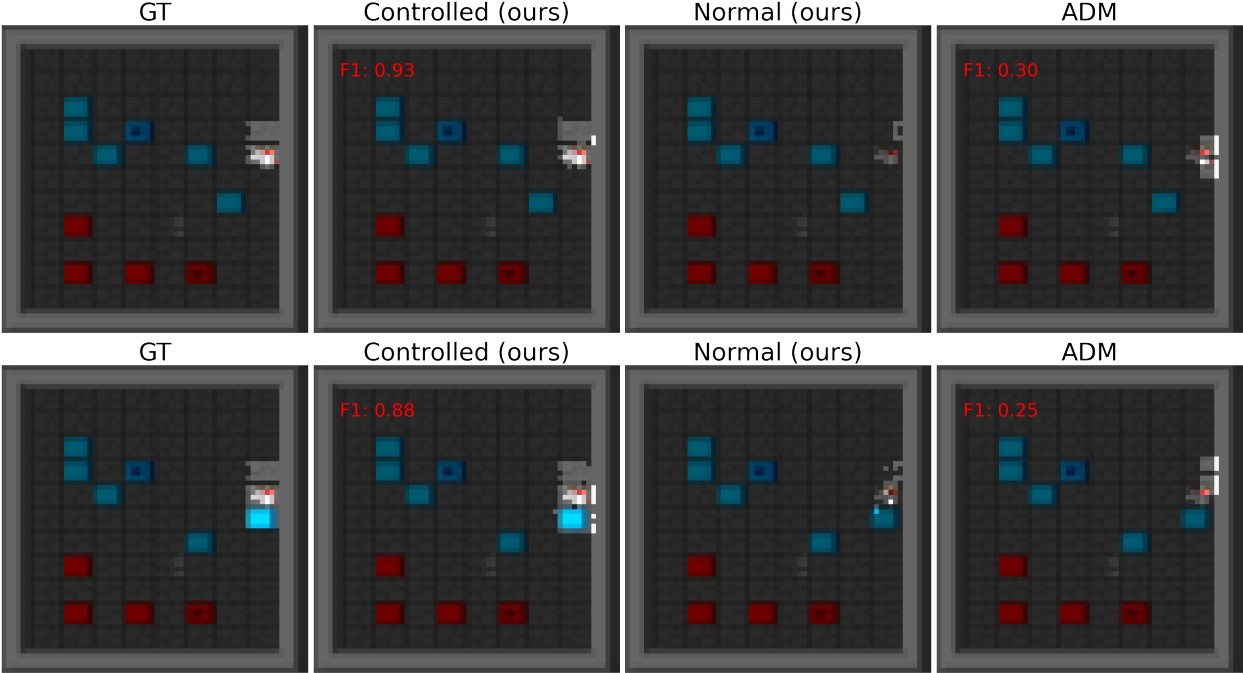

Figure 10: Examples of masks for Clusters.

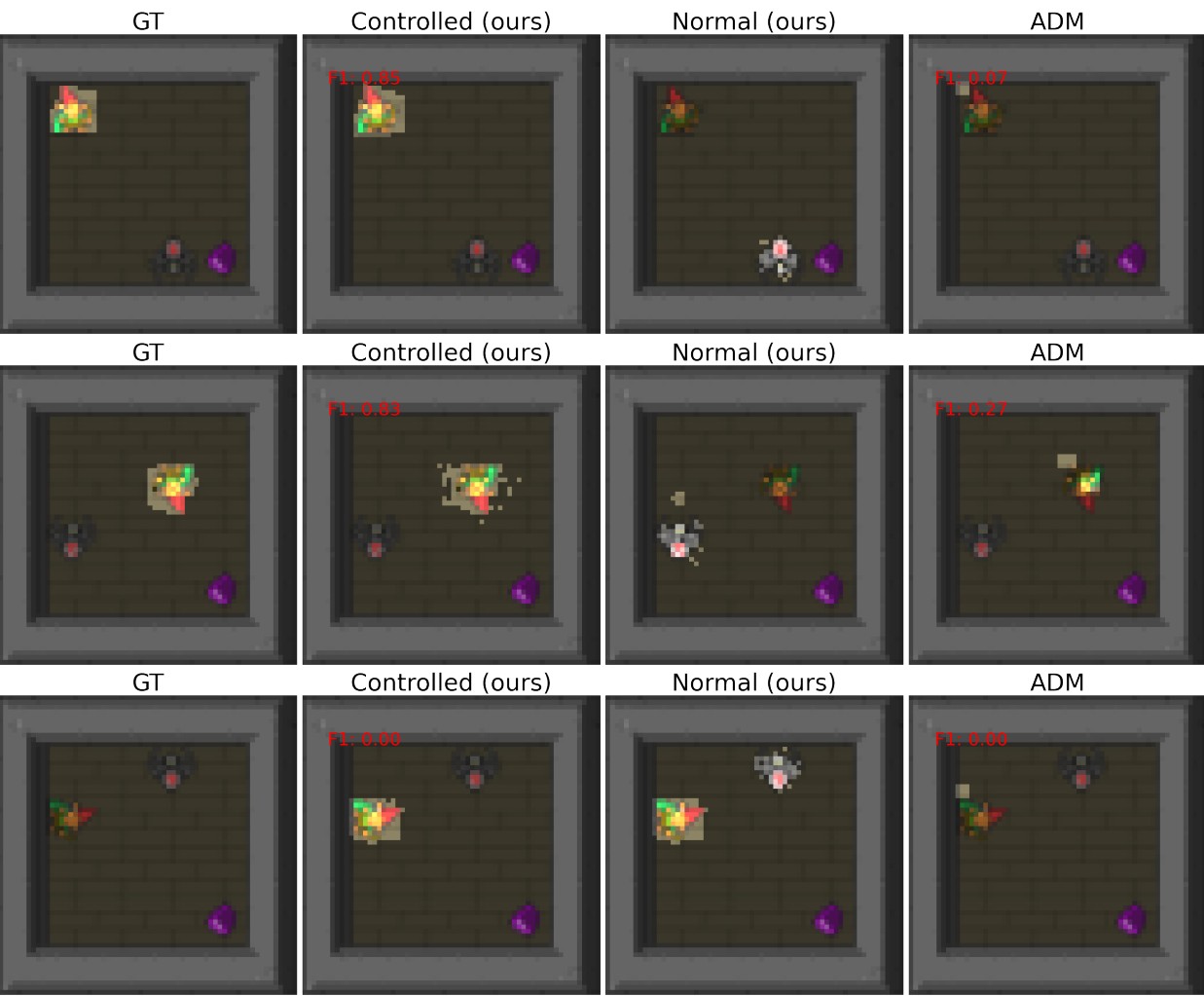

Figure 11: Examples of masks for Spiders.

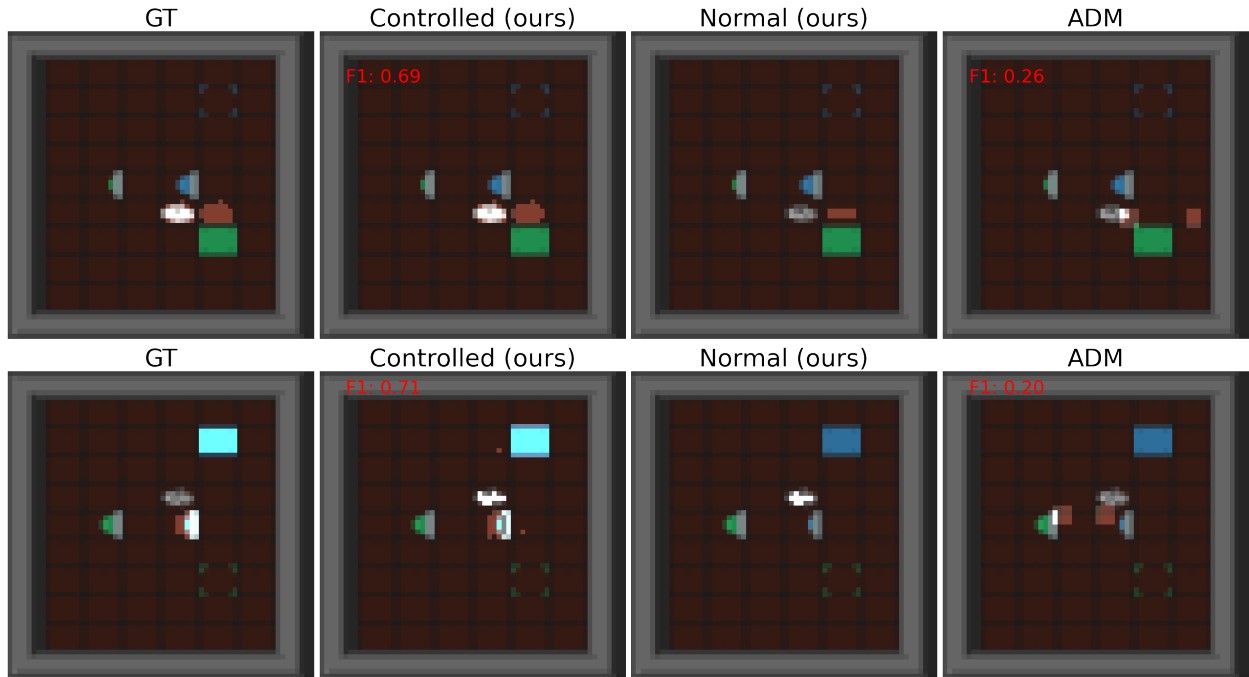

Figure 12: Examples of masks for Lights.

## C    Relation of Blame to the advantage function

A typical use of the formulation introduced in section 2 is to compute the causal effect of an action on the return $G$ relative to a policy $\pi$ as

$$
\begin{aligned}
ICE_G^a(s) &= G^a(s) - \beta_G(s) \\
&= Q(s,a) - V(s) \\
&= A(s,a).
\end{aligned} \tag{6}
$$

$G^a(s)$ is the return the agent would get if action $a$ were to be taken at state $s$ and is typically estimated using a state-action value function $Q(s,a)$. The choice of normality for $\beta_G(s)$ is to estimate the expected return with the state-value function $V(s)$, giving us the advantage function $A(s,a)$. Inspired by Generalized Value Functions (Sutton et al., 2011; Sutton & Barto, 2018) which aim to integrate general knowledge of the world; Blame takes a more general approach than the advantage function by leaving return as special case.

## D    Environments

### D.1    Griddly

Griddly (Bamford et al., 2020) is a highly optimized grid-world based suite of environments. Environments used in this work based on Griddly generate $64 \times 64$ pixel observations, although the size of the grid-world may vary. Griddly supports multiple rendering formats, this work uses the 2D rendering of sprites.

**Spiders:** is a 6x6 arena where a Gnome (the agent) has to grab a Gem without being killed by a Spider. The agent dies if it collides with the spider. In this environment, the agent can move *left*, *right*, *up*, *down* or *stay*. The spider takes an action randomly from the following: rotate left, rotate right or move forward. This environment's controlled entities are: Agent.

**Clusters:** is a 13x10 arena where a Knight has to move boxes of the same color to their corresponding colored-block without touching the spikes. There are two different colors, blue and red. The

agent is rewarded with +1 whenever a box is pushed towards a similar colored block. The agent dies if it collides with spikes or if a box is destroyed by spikes. The agent can move *left*, *right*, *up*, *down* or *stay*. This environment's controlled entities are: Agent and Boxes.

For the RL experiments, we made the environment more sparse by removing all intermediate rewards and only rewarded the agent after all the boxes of the same color are pushed to blocks. Since the agent can not get any reward whenever a box is stuck to a wall. We removed boxes that touches the wall and punished the agent with -0.01. We then scaled the reward of solving a color with the number of boxes pushed to the block. This modifications encourage the agent to solve the environment by pushing the maximum number of boxes into blocks while preventing it from getting deprived from reward by accidentally pushing boxes to walls.

**Lights:** is a 11x8 arena where a Ghost (the agent) has to turn all the lights on by pressing each button. Buttons and lights are colored either blue or green. Pressing a button of one color turns the light of the same color on. The agent can move *left*, *right*, *up*, *down* or *stay*. This environment's controlled entities are: Agent, Buttons and Lights.

**Empty:** this environment is a copy of the clusters environment where all the boxes, blocks, spikes and rewards were removed.

## D.2    Atari Montezuma's Revenge

The ALE (Bellemare et al., 2012a) provides access to Atari 2600 games to learning methods like RL. As it has been a popular choice by methods using inverse models, in this work we use the game Montezuma's Revenge to evaluate CEN. This environment provides uncontrolled as well as controlled effects with more complex entities. The environment typically generates observations of 210x160 pixels which we downscale to 64x64 pixels. Additionally the action space is of size 10.

# E    Training

## E.1    Architecture

**Encoder:** is composed of two 2D convolutional layers with 4x4 kernels, stride 2 and padding of 1. Additionally, we have 2 residual blocks each with two 2D convolutional layers with stride 1 and padding of 1. The first layer has a kernel of 3x3 and the second layer of 1x1. ReLU is used as activation function throughout the network; BatchNorm is used between each layer; and 64 channels on every convolutional layer. We project the resulting maps into a flatten vector of size 32 using a linear layer with ReLU activation function.

**Decoder:** this module is composed of six 2D transposed convolutional layers all having 4x4 kernels, stride 2 and padding of 1. Each layer uses ReLU as activation function but the output layer which uses Tanh activation. Every layer uses 64 channels with the exception of the last layer which outputs a 1 channel prediction of the perceived effects. Parameters are shared among the controlled and normal branch decoders.

**Controlled and normal modules:** both modules are composed of three linear layers with 32 hidden units, each with a ReLU as activation function. The input to the controlled branch are the encoded observation and an embedding of size 8 of the chosen action.

**PPO Agent:** uses an encoder consisting of 3 convolutional layers with (channels, padding, strides) equal to (32, 8, 4), (64, 4, 2), (64, 3, 1) respectively. The encoder is followed by two linear layers of sizes 512 and number of actions respectively, to transform the feature map to the environment's number of actions.

### E.2 Hyperparameters

| Name | Value | Sweep |
|---|---|---|
| hidden size | 32 | [16, 32, 64, 128] |
| latent size | 128 | [16, 32, 64, 128, 256] |
| channels | 64 | [16, 32, 64, 128] |
| learning rate | 0.0001 | [0.0001, 0.0005, 0.001, 0.005] |
| $\alpha$ | 0.01 | [0.001, 0.01, 0.1, 1, 5, 10, 20, 30, 50] |
| $T$ | 0.01 | - |

Table 3: CEN hyperparameter sweeps and final values used.

| Name | Value | Sweep |
|---|---|---|
| entropy | 0.05 | [0.01, 0.05, 0.1, 0.5, 1, 5] |
| hidden size | 64 | [16, 32, 64, 128] |
| attention size | 128 | [32, 64, 128, 256] |
| learning rate | 0.0001 | [0.0001, 0.0005, 0.001, 0.005] |
| $T$ | 0.01 | - |

Table 4: ADM hyperparameter sweeps and final values used.

| Name | Value | Sweep |
|---|---|---|
| encoder channels | 32 | [32, 64] |
| encoder hidden | 32 | [16, 32, 64, 128] |
| latent size | 128 | [64, 128, 256] |
| learning rate | 0.0001 | [0.0001, 0.0005, 0.001, 0.005] |

Table 5: Inverse model hyperparameter sweeps and final values used.

| Name | Value | Sweep |
|---|---|---|
| batch size | 512 | [64, 128, 512, 1024] |
| latent size | 32 | [8, 16, 32] |
| CEN encoder output size | 128 | [16, 32, 64, 128] |
| learning rate | 0.0005 | [0.00005, 0.0001, 0.0002, 0.0005, 0.001] |
| IR Beta | 0.001 | [0.0001, 0.001, 0.002, 0.005, 0.01, 0.1] |
| Rollout size | 2048 | [1024, 2048, 4096] |
| discount | 0.95 | - |
| epochs | 10 | - |

Table 6: PPO hyperparameter sweeps and final values used.

