# OpenReview forum: "Did I do that? Blame as a means to identify controlled effects in reinforcement learning"
_TMLR — Accepted by TMLR_

### Review · Reviewer_utPk · 2022-05-29

**Summary Of Contributions:**

The authors take a causal and counterfactual view on attributing effects to an action for an RL agent.

They introduce a specific neural architecture (the Controlled Effect Networks) for predicting effects (as in model-based RL) with one branch that has access to the action and one that doesn't, as well as an objective function such that the uninformed branch (which they call the normal branch) tries to explain as much as it can, in order to leave the informed branch (which they call the controlled branch) focus on the action-specific changes in observations.



**Broader Impact Concerns:**

None I can foresee besides the general danger of generic advances in AI.


**Requested Changes:**


I believe all of these points can and should be addressed, after which I would probably be satisfied with seeing this paper accepted.
Regarding the last of the above weak points, the issue should at least be discussed, but the paper would be greatly improved if the authors actually ran the modified architecture I am suggesting (which may not make any difference in simple worlds where the effects can be attributed to individual pixels).

- address all the "-" weak points in the above list
- the idea of using linear probes to analyze the internal behavior of deep nets was proposed by Guillaume Alain (2016) "Understanding intermediate layers using linear classifier probes", which should be cited.
- in the list of deep RL + causality papers, I would add Scholkopf et al 2021 "Towards causal representation learning" as well as Ke et al 2021 "Systematic Evaluation of Causal Discovery in Visual Model Based Reinforcement Learning"
- typo: is to us to define --> is for us to define
- typo: to analyzed --> to analyze


**Strengths And Weaknesses:**

Strengths:

+ This paper contributes to a very interesting research question which remains fairly open, introducing a novel way to be more explicit about causal effects in deep RL architectures.
+ The authors draw inspiration from the cognitive science literature, which brings an apparently important point: humans attribute blame in a way that is not just due to the Pearl-style direct causal effect magnitude but also depends on (and contrasts with) what may be considered a normal outcome.
+ The experiments seem to confirm that the proposed approach can do a good job of attributing effects, either in pixel space or in the space of hand-crafted attributes that are known to be relevant to the task, using specially designed learned probes.

Weaknesses:

- The notion of normality is too vague, both in its textual explanations and in the mathematical formalization: I did not understand Eq. 5 clearly enough (what are the different symbols and letters and the expectation standing for exactly?)
- The notion of normality seems to depend heavily on the current policy (which is ok, but should be stated more clearly)
- The statement "Eq. 5 would indicate that no changes to the health bar are normal" on p.3 needs some clarification. The reason (I think) for no change in health being the "normal" outcome is not that one normally does not die from bombs, but rather that the current policy understands the effect of bombs and how to avoid them or not, and thus the learned model should know that the policy will escape the bomb (since it is "easy" to do) and thus the expected effect would be no change in health.
- Adding the normal and controlled branch effects in pixel space does not make sense in general: the addition should be in some appropriate abstract space. Why not add them up in h-space and then apply the decoder?

---

> ### Author Response · Authors · 2022-07-02
> **Response to reviewer utPk**
>
> **1. The notion of normality is too vague...**
> We have added some clarifications that we hope help the reader to understand Blame better.
>
> **2. The notion of normality seems to depend heavily on the current policy (which is ok, but should be stated more clearly**
> We state this in the limitations section.
>
> **3. The statement "Eq. 5 would indicate that no changes to the health bar are normal" on p.3...**
> We have clarified this in the paper, the idea is that on average (over all possible outcomes) health remains the same.
>
> **4. Adding the normal and controlled branch effects in pixel space does not make sense in general...**
> The limitations section mentions that it would be better to model the state in the latent representation using an LSTM or something more sophisticated instead of directly using observations. Regarding the modification in the architecture to add the controlled and normal representations in latent space instead of pixel space, the decoder would not be able to decode controlled effects, which is critical for our experiments. On the other hand, if the downstream task does not need controlled pixels, CEN could be trained as the reviewer said. We have added this distinction to the text in case this is useful for anyone using CEN.
>
> **5. the idea of using linear probes...**
> Thanks, we have added this reference.
>
> **6. in the list of deep RL + causality papers...**
> Thanks, we have added both references.
>
> **7. typo**
> Thanks, changed

---

> > ### Comment · Reviewer_utPk · 2022-07-07
> > **Post-rebuttal comment**
> >
> > I am satisfied with the changes regarding the points I had raised.

---

### Review · Reviewer_4dHp · 2022-06-02

**Summary Of Contributions:**

This paper proposes a causal interpretation for learning affordances. Specifically, the paper proposes a notion of blame, in contrast to individual causal effects (ICE), which measures the difference between what the agent observes versus what the agent expects to happen when no intervention (or action) is taken. Individual causal effects would instead measure the counter-factual difference/effect of taking a different intervention, and really finding if there exists an intervention that would lead to a different outcome.

The paper then applies the definition of blame to a self-supervised network for predicting the effect of an action. The Controlled Effects Network (CEN) tries to disentangle the controlled effects and uncontrolled effects to predict the outcome of the action (i.e. intervention) made by the agent. This is done through the loss function in equation 6 and a two stream architecture where the controlled effects and uncontrolled effects are modeled through separate network streams with a shared base encoder.

The main questions the authors want to address are:
- Can CEN identify controlled effects at the pixel-level?
- Are there applications which don't require pixel-level precision and can CEN predict controlled effects at the attribute level.

While not explicitly stated as a question for empirical study, another claim made in the methods section is: "Since this branch [normal branch] cannot predict everything without the action the model will converge to the expected effect due to the MSE loss."

The paper uses Attentive Dynamics Model (ADM) [Choi et al, 2019] as a baseline for comparisons when answering these questions. Overall, the empirical results confirm that CEN is able to better predict the effects of an action on the environment. The paper also provides some evidence that the two streams are decoupling the controllable and uncontrollable effects.

### Main Contributions
- Connection between blame (as defined by [Halpern, 2016]) and affordances for use in a reinforcement learning/self-supervised learning setting.
- Introduction of Controlled Effects Network (CEN) which is a two stream architecture to predict the effect of an action on the environment, and a novel objective inspired by the previous contribution.
- Empirical experiments focusing on CEN's ability to predict effects in the environment in several domains compared with ADM.
- A discussion about limitations of the approach leading to the potential for future work.


**Requested Changes:**

Overall, I don't think this paper is quite ready for publication. I'm going to break my suggestions into
- Writing/Organization: different ways of organizing the paper to make the concepts clearer, suggestions on how to better present some of the background
- Method: thinking about the core proposed architecture
- Empirical: some suggestions on better ways to test the main hypotheses, and some suggestions on how to make this paper more relevant to the RL literature.


## Writing/Organization

These are all suggestions aimed at making the paper clearer and focusing in on the key concepts presented.

WO-1. The paper focuses a lot on the difference between ICE and blame. I'm wondering if section 2 focused more on blame and presented ICE as some auxiliary. I think it is reasonable to motivate moving away from ICE, but I think spending so much time on the idea is drawing focus and attention away from the actual proposal.

WO-2. I'm confused why you define the advantage function in terms of blame as this isn't used anywhere else in the paper.

WO-3. You should cite [Horde: A Scalable Real-time Architecture for Learning Knowledge from Unsupervised Sensorimotor Interaction][https://aamas.csc.liv.ac.uk/Proceedings/aamas2011/papers/A6_R70.pdf] when discussing GVFs, not the RL textbook.

WO-4. I'm wondering what the authors desired core proposal is. The initial sections read more like a larger discussion on using blame as a core concept in reinforcement learning that can be used to learn affordances but also explain other RL concepts. The rest of the paper is an architecture proposal, and doesn't really have much connection to RL beyond needing agent's which take actions. I think if the authors can clarify this (i.e. focus on the architecture proposal) it would clean up a lot of the unneeded text and make the paper much easier to read/follow.

WO-5. I think the idea of blame, or the difference between what we expect and what is observed, is a useful concept for RL, but I feel the paper doesn't do a good job bridging between the two concepts well. I think if the paper instead proposed ICE and blame using more RL notation/language the idea would be more digestible for an RL audience (which I interpret as the main audience of this paper).

WO-6. The empirical section would be more complete if details on the ground truth used was included. You have space.

WO-7. I would like more context about the experiment in the figure captions. Also I think the plots axes should be labeled (currently there are very few labels for any of the axes)

## Questions about Methods/Concepts
This section is mostly questions/clarifications on some of the concepts.

Q-1. What is the connection between equation 5 (i.e. blame) and general value functions? If there is a connection the paper should make this clearer.

Q-2. Equation 3 suggests the notion of blame used in this paper is equivalent to ICE defined in equation 1. Is this actually the case? This might also be because the definition of ICE in equation 1 is not clear. My interpretation is that it returns a boolean on whether there exists an intervention $\tilde{x}$ which changes the observed outcome for individual $Y_i$. This seems quite different from measuring the difference between what we expect to happen to individual $Y$ (i.e. normality through $\beta_Y$) and the observed outcome over intervention $x$.

Q-3. I'm not sure I understand the connection to the dueling networks architecture. While there are two stream in that architecture the goal of the two streams in Wang et al's architecture is to estimate the value function and the advantage function separately. Your architecture separates between controllable and uncontrollable dynamics in the environment. So maybe you meant just in the sense that there are "two streams" in the architecture?

Q-4. How would this be applied in a reinforcement learning agent? Would you use the learned representations to learn policies, would you use the bit masks to drive attention in the agent? I think the paper leaves this unclear.

## Empirical Section

### E-1
I think the core questions addressed by the authors in their empirical exploration is mismatched from the claims made throughout the paper about their approach. While I think there are several questions that need addressed in a better way, the main missing claim the authors seem to miss in their testing is the claim made in section 3:

    Since the this branch [normal branch] cannon predict everything without the action the model will converge to the expected effect due to the MSE loss.

I think the best way to test this question is the inclusion of more baselines. I suggest two new architectures to explore this claim:
1. Same objective function, but combine the streams into a single stream with only two linear heads on the output to learn $e^a_c$ and $e^n$.
2. Same objective function and architecture, except include the action into the normal stream as well as the controlled stream.

These two architectures should address the thrust of this claim adequately. The first gives direct insight into whether the separation of streams improves performance and the second tries to get at the responsibility of the action on the possible parts of the dynamics modeled by each stream.

In the end, it would be even better to directly measure what parts of the dynamics are being modeled by the two streams. I would make a suggestion here, but I think the current measurement is too unclear for me to make a reasonable suggestion in the current framing of the paper. You begin to evaluate this on Montezuma's Revenge by showing the masks for the different streams, but I would like to see this go further.

I also think the mixture enabled by the hyperparameter $\alpha$ should be added to the main work. Specifically, the values between [0, 1]. Another interesting direction here is to have the two parts of the objective be a true mixture (i.e. $L = (1 - \alpha) MSE(e_c^a + e_n, e^a_p) + \alpha MSE(e_n, e_p^a)$). This would enable a better understanding of what each part contributes to the overall performance of the proposed architecture.

### E-2

I don't quite understand how the bitmasks are constructed for both CEN and ADM. I think this procedure and how the threshold is chosen should be expanded in the main paper. I also found the explanations in the appendix to be unsatisfactory.


### E-3

While I'm not sure it is required for publication (I think the above issues are already quite a bit), applying your approach in some of the same experiments proposed by ADM would make the comparison more interesting generally. It would also help narrow in on the implied hypothesis of this paper (more explicitly stated in the ADM paper): "better separation of controllable and uncontrollable effects in the agent's dynamics learning will lead to better behavior". This paper and the ADM paper share this hypothesis and if you can better show your method does indeed separate the effects and if the final agent performs better on the given tasks this can be evidence in favor of the question. Again, I'm not sure it is needed for publication at TMLR, but would make the paper significantly stronger.


## Missing citations/connections

While I don't give a full account of missing works, I'll give some lines of research I think is relevant to the work presented in this paper.

- Affordances: https://proceedings.mlr.press/v119/khetarpal20a.html, https://journals.sagepub.com/doi/abs/10.1177/1059712321999421?casa_token=fFj0p4elcUQAAAAA:nyXqZgmomqT68jPdlauSIKgOkJyqBKZytGwFIf0Tkxw_BaRxgCu3q70MLuhc6Xp7MJJOeoV1ot8V3g
- Model-based reinforcement learning: Generally, I think this idea is related to model based reinforcement learning (as you are trying to model the effect of an action on the environment). While this may be applied in other aspects of the agent, generally this kind of modeling is discussed quite a bit in MBRL. https://proceedings.neurips.cc/paper/2015/hash/6ba3af5d7b2790e73f0de32e5c8c1798-Abstract.html, https://arxiv.org/abs/2006.16712, etc...
- General Value Functions (if applicable): https://aamas.csc.liv.ac.uk/Proceedings/aamas2011/papers/A6_R70.pdf, https://journals.sagepub.com/doi/abs/10.1177/1059712313511648?casa_token=ttGynWjW6OwAAAAA:t1XDAZQv9K2Lm7UPrAaJW5rYK3SgLMvJEXMM0x71DRvHL5KZLmQdKunSfO83x0j049V9hXsj5ix8qQ, etc...

## Minor Suggestions/edits:
- Section 2.2: "actual causality" -> You should cite where you are getting the definition of normality from.
- What is the p in e^a_p in equation 2? If it stands for "perceived" is it really needed in this context as we aren't discussing the actual underlying effect.
- You mention three main questions at the beginning of section 4, but only list one question and then conjecture about what might be needed in terms of precision.

**Strengths And Weaknesses:**


### Strengths
- Clear acknowledgment of limitations
- Simple idea leading to empirical improvements in the results provided

### Weaknesses
- Limited set of baselines to accurately test the core hypotheses.
- Unclear where this fits into an RL agent. The paper lightly suggests the disentangling these effects could be useful to RL agents, but conceptually where this sits in an agent isn't clear to me given the text of the paper.
- There are parts of the paper that seem unnecessary given the core proposal/hypotheses (see below in organization).
- Much of the motivation seems centered on the fact that this is "more human like". I find this kind of motivation a bit hard to digest given our current understanding in the cognitive sciences. I think the core concept of blame the paper proposes can stand independently from this line of motivation.

---

> ### Author Response · Authors · 2022-07-02
> **Response to reviewer 4dHp**
>
> **WO-1. The paper focuses a lot on the difference between ICE and blame...**
> Although it is true that Blame could be understood without much details about ICE, we believe that this gives a useful overview and justification to why use Blame instead of more traditional causal measures. This may be particularly interesting to researchers from the causal community.
>
> **WO-2. I'm confused why you define the advantage function...**
> This is now in the appendix since we think is an interesting relation.
>
> **WO-3. You should cite Horde...**
> Thanks, we have cited the Horde paper.
>
> **WO-4. I'm wondering what the authors desired core proposal is...**
> We have added an RL experiment to showcase CEN as intrinsic motivator and how identifying controlled effects can aid exploration.
>
> **WO-6. The empirical section would be more complete if details on the ground truth used..**
> This information is in the appendix but we have now moved it to the main text for clarity.
>
> **Q-1. What is the connection between equation 5...**
> We have moved this part to the appendix. The main idea is that Blame, as GVF, departs from reward to model generic features of the environment.
>
> **Q-2. Equation 3 suggests the notion of blame used in this paper is equivalent to ICE defined in equation 1...**
> Eq. 3 uses the magnitude and direction of ICE but Eq. 1 does not. This mismatch is because ICE is typically defined as inequality in causal literature (we only want to know if there is a causal effect or not) but for RL the magnitude and direction are important. We have changed the text to be consistent and use magnitude and direction everywhere.
>
> **Q-3. I'm not sure I understand the connection to the dueling networks architecture...**
> The action stream of the dueling networks can be understood as the controlled part of the return by action $a$, i.e. how much would the return change because of this action. The value stream is the return the agent would normally get.
>
> **Q-4.} How would this be applied in a reinforcement learning agent?**
> We have added some ideas to the conclusions section and also showcased CEN as intrinsic motivator in one of our experiments.
>
> **E-1 I think the core questions addressed...**
> Regarding the convergence of the normal branch to the expected effect due to the MSE loss, we cannot test if this is true since we do not know what is the average effect in the first place. The intuition why the normal branch should converge to the expected effect is because without knowing the action, the best prediction the model can have is the weighted (by the action probs.) mean effect. Regarding the first proposal, it would be extremely hard (impossible?) for the network to create a representation that is linearly separable into controlled and normal effects. Even if the network could, it is not clear to us how this shows that the normal branch is converging to the expected effect. Regarding the second proposal, including the action would make the model an overparametrized forward model which we know can predict effects and next observations. We believe that our ablation experiment on the alpha hyperparameter provides evidence of why the action is needed since for high alpha the normal branch tries to model everything that could change. Note that we provide normal and controlled masks for all the environments in the appendix. We added the ablation experiment to the main text.
>
> **E-2 I don't quite understand how the bitmasks are constructed...**
> This is now in the main text. The threshold was chosen empirically for both CEN and ADM to maximize their performance in each environment.
>
> **E-3 While I'm not sure it is required for publication...**
> Unfortunately, ADM's count-based method has not been open-sourced. We have added an additional experiment using Never Give Up (current SOTA on exploration) where we replaced the inverse model used to identify what is controlled with CEN.

---

> > ### Comment · Reviewer_4dHp · 2022-07-06
> > **Thanks for the extra comments and details!**
> >
> > WO - All these edits look good to me! Thanks!
> >
> > Q1 - While this is true, GVFs are modeling the general observation variables in a TD/value function formulation rather than a more supervised formulation. While I think the bridge is ok, I think the distinction wasn't quite apparent here.
> >
> > Qs - thanks for the other clarifications!
> >
> > E1. I agree that the ablation study helps here. How many runs and are there any error bands reported in this plot? Hard to draw conclusions if there is a single run.
> >
> > While I can accept some of the reasoning/intuition behind not running my proposals, I still believe they would help distinguish your model and the choices you make. The goal of the proposals was not to have something working as well as yours, but to check the intuitions you used to construct your model.
> >
> > E2. Thank you for the clarification and addition to the main text. I think this is definitely important to be in the main text.
> >
> > E3. I'm unsatisfied with this reasoning for not including an important baseline. Did you try to implement the strategy proposed by Choi et al? If so did it just not work quite right? Did you contact Choi to see if they have the code they can share?
> >
> > Some more comments:
> >
> > C1 - I think there is another limitation of your work, not shared by Choi et al. This would be the need for ground truth labels. From what I understand ADM does not require this information. This should be highlighted, as in effect your method is given more information to work with as compared to ADM. I may be mistaken, so if ADM shares this you should also highlight this as a core limitation of yours and their work.
> >
> > E4 - In the last paragraph of page 5 (talking about the special properties of the MZR API), you discuss reseting the environment to learn the causal effects of a particular action. This is a heavy limitation of the empirical evaluations in my opinion. Do all your empirical evaluations share this procedure? Did you test your method with out being able to take advantage of this property of the environment? Did the other baselines have similar abilities/more data to compensate for this procedure? If this is required by your algorithm, do you have a good motivating example (outside of a simulated environment like atari), which we can apply this kind of procedure?
> >
> > E5 - The new experiments using your method for intrinsic motivation just emphasizes the missing baselines from Choi et al. I really think these comparisons are necessary now, especially with the added information your model gets about the data stream (i.e. ground truth labels, ability to restart the env to learn the normal effect, etc...).

---

> > > ### Author Response · Authors · 2022-07-07
> > > **Follow up**
> > >
> > > > **Thanks for the quick response, we really appreciate this feedback.**
> > >
> > > **C1 - I think there is another limitation of your work, not shared by Choi et al...**
> > >
> > > There seems to be a misunderstanding with the use of ground truth. CEN does NOT need nor use the ground truth, it is a fully unsupervised method. CEN only needs an observation and action as input.
> > >
> > > Moreover, due to the fundamental problem of causal inference, such a ground truth does not exist in general (we cannot go back in time to try another action). We can compute ground truth for the environments used since we know the world's dynamics. This ground truth is only used to evaluate CEN and baseline methods, in other words, the ground truth is used in the same way for all the methods in this paper, thus there is no advantage for CEN in any way.
> > >
> > > **E4 - In the last paragraph of page 5 (talking about the special properties of the MZR API)...**
> > >
> > > As mentioned in C1, this is done to get ground truth so we can evaluate the methods (CEN, ADM, and the Inverse model), not for training nor finetuning of any kind. We have added this clarification to the main text to avoid confusion.
> > >
> > > **E3. I'm unsatisfied with this reasoning for not including an important baseline...
> > > E5 - The new experiments using your method for intrinsic motivation...**
> > >
> > > See the answer to C1 regarding the misunderstanding on ground truth and restarts (CEN does not use this additional information). Regarding the RL experiments, we compare CEN to the inverse model used in Never Give Up (NGU), which is a stronger baseline than ADM based on two independent evaluations: table 1 in this paper and the Atari results in the NGU paper. However, we could reimplement Choi's count-based method or even other clustering methods for RL exploration but this is far from the main goal of this work. Clustering of states for RL is an important research direction that goes beyond causality and controllability. Furthermore, our experiments using CEN as intrinsic motivator are there to showcase the use of controlled effects but there are many other use cases, like hierarchical learning. We leave the exploration of these many possibilities to the research community if deemed interesting.

---

> > > > ### Comment · Reviewer_4dHp · 2022-07-08
> > > > **Thanks**
> > > >
> > > > I'm reasonably satisfied with the clarifications provided by the authors! I think the additions and clarifications to the main text make the paper much stronger and more clear.
> > > >
> > > > Thanks!

---

### Review · Reviewer_b3Ck · 2022-06-08

**Summary Of Contributions:**

The authors introduce a novel representation learning scheme inspired by psychological theories of causal reasoning in humans. Specifically, they assess causal influence by comparing potential interventions to the "normal" dynamics. Dimensions of the observation space (pixels for all experiments) that differ are considered to be controlled by the agent.

To evaluate this notion of causal influence, a forward model is learned with a unique architecture that forces separate interventional and "normal" streams whose predictions are subsequently dimension-wise summed to produce the final step-observation prediction. An auxiliary loss is used to prevent the normal stream from degenerating: its log likelihood is maximized independent of the interventional streams contribution. This modeling approach is called the Controlled Effect Network (CEN) and its efficacy in determining which pixels are controlled is evaluated experimentally.

**Broader Impact Concerns:**

No broader impact statement given, but I don't think one is needed here, as all possible concerns would be shared with the whole subfield of Deep RL.

**Requested Changes:**

1) A good deal of the motivation for this work appears to stem from recent advances in exploration. While CEN does do similar things to methods that have been shown to aid in exploration, no direct evidence of CEN's exploratory utility is given in this paper. Either some experimental results along these lines should be attempted, or this connection should be limited and CEN's utility for exploration explicitly noted as an open question.

2) The probe experiments shown in Table 1 are interesting, but two additional baselines would better help me contextualize CEN's performance. If I understand correctly, Probe accuracy from pixels should be highest (for both controlled and uncontrolled attributes) for unmasked images. Could you report these numbers? This would confirm the relative predictability of each attribute; I worry the uncontrolled attributes might just be harder to predict regardless of the mask. Similarly, probe accuracy from CEN's latent representation prior to training (i.e. random weights) would shed light on the contribution of inductive biases relative to objective functions.

3) A few connections to related work are probably worth including (Specific papers are merely illustrative, feel free to cite different papers if you feel they make more sense in addressing the following points):
* The notion of "normality" feels quite similar to that of "passive dynamics" from control theory. I couldn't track down the original citation for this, but "Efficient computation of optimal actions" by Emo Todorov is fairly representative of its usage in the field.
* The discussion around the limitations of inverse models is solely focused on single-step inverse models. But many recent works explicitly deal with multi-step predictability in either the form of action sequences (e.g. Mohamed & Rezende, 2015 -- already cited for a different reason) or options (e.g. "Variational Intrinsic Control"). It's probably out-of-scope to compare to such methods since it's unclear how best to adapt them to produce segmentation masks, but their existence should mentioned and subsequently hedge some of your conclusions about the superiority of CEN to inverse modeling approaches.

4) The relationship between the RL notion of Advantage and your notion of "Blame" is interesting if this hasn't been previously published elsewhere (its new to me at least), but it doesn't seem particularly relevant to the rest of the paper. Particularly since "Blame" is only calculated locally, rather than being the expected value of some future quantity, no reinforcement learning is going on, etc. Assuming this is novel, then I believe it'd be better served in the appendix (perhaps a "relationship between blame and rl" section?). All of this goes double for the GVF reference -- its not transparent how Eq 5 relates GVFs, but its unclear how relevant it'd be to the rest of the paper even if it were. So I'd suggest unpacking the relationship in the appendix. (Could also just cut out these bits altogether, but it is an interesting connection.)

5) As mentioned in limitations, using an expectation over pixel values for the "normal" stream feels very limiting, please explicitly address or acknowledge this.

**Strengths And Weaknesses:**

Strengths:
Overall, the work is very easy to follow. Particularly in the context of work at the intersection of psychology, causality, and reinforcement learning, which tends to yield either impenetrable jargon or hand-waving with respect to at least of one of the fields involved.

The method is simple, novel, well-motivated, and supported by a reasonable set of experimental results.

I know the limitations and requested changes are longer than this strengths section, but I promise you that these strengths outweigh the rest :)

Limitations:
Most of the obvious limitations are well-addressed by the paper. Using a random policy for "normality" feels particularly limiting in the context of large/high-dimensional action spaces (e.g. humanoid or robotic hand), but this is fine as the first step in a novel direction of research.

That said, I was quite surprised the expectation in Eq 5 wasn't discussed more. This expectation over observations could easily backfire even in very simple environments. For example if action_0 shifts a certain observation dimension by -1 and action_1 shifts the same dimension by +1, then the "normal" stream will predict no change for that dimension, even though change always occurs. Dealing with the full distribution of observations under the "normal" policy seems more ideal. Something like the KL between e_p^a and E_\hat{a} e_p^\hat{a} or some other measure between distributions (e.g. Wasserstein). Obviously, this would be much harder implement, but talking about the ideal case and then showing that the expectation over observations is "good enough" would be sufficient -- this limitation should be mentioned explicitly.

---

> ### Author Response · Authors · 2022-07-02
> **Response to reviewer b3Ck**
>
> **0. That said, I was quite surprised the expectation in Eq 5 wasn't discussed more...**
> In the given example, as the reviewer said, the normal stream will predict 0. Then action\_0 would have an effect of -1 and action\_1 an effect of +1. We do mention in the limitations section that our choice of normality is not perfect and that more research needs to be done. We have added more details to Eq 5 and extended our discussion in limitations to propose to use a critic as in GANs to approximate the mode not the mean.
>
> **1. A good deal of the motivation for this work appears to stem from recent advances in exploration...**
> We have added two experiments that showcase the potential use of CEN as intrinsic motivator to improve exploration and compare it to the inverse model used in the state-of-the-art exploration agent Never Give Up.
>
> **2. The probe experiments shown in Table 1 are interesting, but two additional baselines would better help me contextualize CEN's performance...**
> These baselines make a lot of sense, we have added them to the appendix to not clutter the current table. We have also included an additional experiment for probing using the normal hidden representation with random weight initialization. These two last experiments show that CEN's access to the action (in contrast to action-prediction methods) makes the task much easier for environments without any uncontrolled dynamics (Clusters and Lights). It is known that even random weights produce a decent lower-dimensional representation of the input, which in this case encodes the action making it easier to solve the task. More details in the experiment's results.
>
> **3. A few connections to related work...**
> Thanks for this, we were not aware of the relation to Todorov's work. We have added it to the related work section. We have also added the work on multi-step control.
>
> **4. The relationship between the RL notion of Advantage and your notion of "Blame"...**
> As far as we know, this link is novel. We do agree that it may make the understanding of Blame more confusing. We have moved the link to the advantage function and GVFs to the Appendix.
>
> **5. As mentioned in limitations, using an expectation over pixel values for the "normal" stream...**
> What kind of change would you be expecting? We do acknowledge this limitation in the limitations section.

---

> > ### Comment · Reviewer_b3Ck · 2022-07-07
> > **Thank you for your revisions, but issues remain.**
> >
> > Let's consider issues 0,3,4, and 5 to be resolved. I greatly appreciate these revisions.
> >
> > ## Issue 6:
> >
> > I apologize for not initially catching this, but you've highlighted results in Table 1 that have strongly overlapping confidence intervals (e.g. all of the MR results). This isn't recommended and weakens the case for your interpretation of these results.
> >
> > ## Issue 2:
> >
> > The new probe experiments are quite interesting. I'm a bit surprised how low the spider and skull predictions are. Having those be low for CEN is taken in the main text to be strong support for your hypothesis (i.e. it learns to ignore uncontrollable elements). But now it's clear that CEN-rand differentially ignores uncontrollable elements without any learning. It feels like this result should significantly change the narrative of the paper e.g. the architecture provides a strong inductive bias towards modelling controllable elements.
> >
> > ## Issue 1:
> > The new exploration experiments feel quite weak to me. The NGU paper (Figure 2) shows results that suggest inverse modelling representations are sufficient for episodic coverage. You show that this isn't the case in a far simpler setting -- why would their method fail here and not there? One explanation might come down to the evaluation metric. You use a state visitation map whereas they use a quantitative measure of coverage (percent of states visited per episode). Is the state visitation map episodic or lifetime? I think reusing NGU's metric makes more sense here regardless.
> >
> > Another cause for concern is that neither algorithm actually results in uniform state coverage in the simple simple environment. Surely this should be possible even with the base state representation (one-hot), so I'm really not sure what I should be taking away from this experiment.
> >
> > The second new experiment looks more straightforward, but the simplicity of the problem and short training time suggest that architectural biases might account for most of this gain (particularly given issue 2). You use Montezuma's Revenge as the flagship example throughout the rest of the paper. If you want to claim that your representation improves exploration, you should demonstrate it on this well-studied exploration problem.
> >
> > I'm afraid I can't vote to accept this paper in it's present state. As fixing the exploration experiments would take significant time, I'd suggest cutting these along with all of the textual connections to exploration and relegate this to the discussion section / future work.

---

> > > ### Author Response · Authors · 2022-07-07
> > > **Response**
> > >
> > > > **Thanks for the quick response, these comments helped us improve the paper**
> > >
> > > **Issue 6: I apologize for not initially catching thi...**
> > >
> > > Is the reviewer referring to the first block (pixels) experiments? The results for the second block (latent) do not seem to be overlapping. If this is the case, it is true that ADM has high variability and sometimes performs similarly to CEN. We have relaxed our claim in section 4.2.1.
> > >
> > > If we misunderstood this issue, we would welcome a clarification.
> > >
> > > **Issue 2: The new probe experiments are quite interesting...**
> > >
> > > In environments without uncontrolled effects (Clusters and Lights), action and agent are correlated which makes easier the task of the linear probe. If the relation between object and action is more complex (e.g. boxes, lights or buttons), the linear probe has more difficulties. This indicates that the architectural biases are enough for simple relations but the more complex the relation the more learning is needed.
> > >
> > > In environments with uncontrolled effects, CEN learns to model controlled objects while still ignoring uncontrolled elements. The main text points out that only having high score on controlled objects is not enough since the model could also represent both controlled and uncontrolled. What we want is high score on controlled objects but low score on uncontrolled objects. We do have that with an untrained CEN but clearly not as good as with a trained one. The main factor seems to be the relation between action and object, if this is a simple relation the linear probe can do well enough but in more complex relations learning becomes more important.
> > >
> > > Having said that, we believe these architectural biases are, too, part of this work's proposal. It is not trivial to disentangle effects while having access to the action, that's why most research uses action-prediction models.
> > >
> > > **Issue 1: The new exploration experiments feel quite weak to me...**
> > >
> > > The reason for the discrepancy between NGU Figure 2 and CEN's Figure 7a, is that NGU with an inverse model bias the agent towards walls. More precisely towards objects, where predicting the action is difficult or impossible. In a wall, the model cannot predict if the action taken was to move towards the wall (e.g. forward) or action without an effect (e.g. attacking or do-nothing). Figure 7a shows how NGU with inverse model is more often near walls. We hypothesize that this may be the reason for the high reward in NGU's Fig. 2 maze, where visiting new walls may provide a higher reward. The map in Figure 7a is the episodic visitation.
> > >
> > > **Another cause for concern is that neither algorithm...**
> > >
> > > As mentioned above, the inverse model is biased towards objects with unpredictable actions (e.g. walls), this makes the trajectory of the agent non-uniform. CEN's agent has a more uniform exploration but it is still not perfect, we believe there are a number of reasons for this. First, Eq. 5 (4 in the updated version) introduces a bias by using a policy to estimate the normal world, see limitation 3. Second, the episodic memory used to provide intrinsic rewards is not perfect and different states may be clustered together.
> > >
> > > **The second new experiment looks more straightforward...**
> > >
> > > As with Issue 2, we believe these architectural biases are, too, part of this work's proposal. It is not trivial to disentangle effects while having access to the action, that's why most research uses action-prediction models. In the same way, an inverse model provides architectural biases that make it work in a certain way that may be beneficial for some tasks.
> > >
> > > MZR is hard to solve without the lifetime (RND) exploration bonus, see NGU ablations. Moreover, MZR does not have that many controllable entities in comparison to Clusters, which is what this paper focuses on. Neither MZR nor Atari are ideal benchmarks, as shown in AMIGO (Campero et al. 2021) methods that perform well in MZR fail in what may seem a simpler task like MiniGrid. We think Clusters provides a cleaner testbed for the identification of controlled effects and help to isolate the gains provided by learning to control the environment.
> > >
> > > **I'm afraid I can't vote to accept this paper in it's present state...**
> > >
> > > Experiments in section 4.3 aim to showcase CEN in an RL setting with sparse rewards where solving the task requires to control objects. Although this is not the main contribution of the paper, we agree that more environments in Atari and Minigrid would make a stronger case for CEN as an exploration tool. Nonetheless, we believe these findings are useful to the research community working on exploration, thus we would wish to leave these results in the paper. Having said that, if the reviewer still thinks these results are of no value to the research community or that they should be left outside the paper, we can remove them from the manuscript as it originally was.

---

> > > > ### Comment · Reviewer_b3Ck · 2022-07-07
> > > > **Response 2**
> > > >
> > > > Let's consider issue 2 resolved, so long as the role of architecture is mentioned in the text and the spider/skull results downplayed/interpretation hedged.
> > > >
> > > > > Is the reviewer referring to the first block (pixels) experiments? The results for the second block (latent) do not seem to be overlapping. If this is the case, it is true that ADM has high variability and sometimes performs similarly to CEN. We have relaxed our claim in section 4.2.1.
> > > > If we misunderstood this issue, we would welcome a clarification.
> > > >
> > > > In Figure 1, all results in the MZR row have overlapping confidence intervals. The CEN and baseline (Inverse) even have an identical (!) F1 Latent values (.61), and yet you've bolded CEN here. Figure 1 should be corrected such that nothing is bolded when there are overlapping intervals, and the text corrected in light of this more tentative evidence.
> > > >
> > > > >The reason for the discrepancy between NGU Figure 2 and CEN's Figure 7a, is that NGU with an inverse model bias the agent towards walls. More precisely towards objects, where predicting the action is difficult or impossible.
> > > >
> > > > This would make sense if the reward was based on model-error, but it's not. It's based on nearest neighbour distance in latent space. So long as each state is mapped to a unique representation, the optimal policy should still eventually cover the space.
> > > >
> > > > > We hypothesize that this may be the reason for the high reward in NGU's Fig. 2 maze, where visiting new walls may provide a higher reward.
> > > >
> > > > The metric (not reward) being plotted in NGU's Fig 2 is the percentage of states visited. As such, this explanation doesn't make sense.
> > > >
> > > > > MZR is hard to solve without the lifetime (RND) exploration bonus, see NGU ablations. Moreover, MZR does not have that many controllable entities in comparison to Clusters, which is what this paper focuses on. Neither MZR nor Atari are ideal benchmarks, as shown in AMIGO (Campero et al. 2021) methods that perform well in MZR fail in what may seem a simpler task like MiniGrid. We think Clusters provides a cleaner testbed for the identification of controlled effects and help to isolate the gains provided by learning to control the environment.
> > > >
> > > > I agree that no benchmark is perfect. But the two primary methods you've compared to throughout (ADM and NGU) both show results on MZR (and many other hard exploration Atari games), whereas neither uses Clusters (indeed, I can't find evidence for Clusters ever previously being used as an exploration benchmark).
> > > >
> > > > > Nonetheless, we believe these findings are useful to the research community working on exploration, thus we would wish to leave these results in the paper. Having said that, if the reviewer still thinks these results are of no value to the research community or that they should be left outside the paper, we can remove them from the manuscript as it originally was.
> > > >
> > > > As they stand, I don't believe these experiments provide significant evidence of CEN's advantages over existing methods for the purposes of exploration. I'd be happy with them being included in the appendix so long as its made clear that they are preliminary and all claims regarding CEN for exploration are abandoned.

---

> > > > > ### Author Response · Authors · 2022-07-07
> > > > > **Response**
> > > > >
> > > > > **Issue 6:** we have removed any bold indicator to avoid confusion and the text describes the results obtained. Our goal was not to show that CEN is 1% better or 10% better but that Blame is an interesting direction to explore further.
> > > > >
> > > > > **Issue 1:** we have moved the experiment to the appendix and (we believe) removed all references to it from the main text.

---

> > > > > > ### Comment · Reviewer_b3Ck · 2022-07-11
> > > > > > **Thanks for the changes!**
> > > > > >
> > > > > > I think all of my concerns have now been reasonably addressed.